# PARETO POLICY ADAPTATION

**Panagiotis Kyriakis**
University of Southern California
Los Angeles, USA
`pkyriaki@usc.edu`

**Jyotirmoy V. Deshmukh**
University of Southern California
Los Angeles, USA
`jdeshmuk@usc.edu`

**Paul Bogdan**
University of Southern California
Los Angeles, USA
`pbogdan@usc.edu`

## ABSTRACT

We present a policy gradient method for Multi-Objective Reinforcement Learning under unknown, linear preferences. By enforcing Pareto stationarity, a first-order condition for Pareto optimality, we are able to design a simple policy gradient algorithm that approximates the Pareto front and infers the unknown preferences. Our method relies on a projected gradient descent solver that identifies common ascent directions for all objectives. Leveraging the solution of that solver, we introduce *Pareto Policy Adaptation* (*PPA*), a loss function that adapts the policy to be optimal with respect to any distribution over preferences. PPA uses implicit differentiation to back-propagate the loss gradient bypassing the operations of the projected gradient descent solver. Our approach is straightforward, easy to implement and can be used with all existing policy gradient and actor-critic methods. We evaluate our method in a series of reinforcement learning tasks.

## 1 INTRODUCTION

Deep reinforcement learning has a pivotal role in solving several control problems of practical interest in multi-agent autonomy (1), robotics, and cyber-physical systems (2) that would otherwise be intractable, such as robotic locomotion (3), the Atari (4) and Go games (5) to name a few. While such approaches have focused on the scalar reward setting, there exist many real-world problems that have multiple, conflicting objectives and, therefore, cannot be addressed by the current reinforcement learning tools. In such scenario, it is challenging to find an optimal control policy as the trade-offs (*preferences*) among the objectives may not be precisely known at training time or may differ from user to user. The paradigm of Multi-Objective Reinforcement Learning (MORL) provides a generalized framework for dealing with multi-dimensional rewards signals and has been identified as one of the main challenges of real-world reinforcement learning (6). However, learning optimal policies for MORL has been proven to be quite challenging, because most strategies require either to have access to or be able to infer a quantification of the relative importance of the objectives, or they perform sophisticated searches in the value or policy space aiming to find an ensemble of policies that are non-inferior to each other. The former methods lack adaptability and generalizability, as their performance is tied to the given or inferred preferences, while the later ones suffer from scalability issues, because they learn multiple policies, and are usually complicated to implement. In this work, we aim to strike a balance between those two families of methods.

**Related Work**: Multi-Objective Optimization (MOO) provides the fundamental tools for MORL. In MOO, there are two common solution concepts: *scalarization* and *Pareto optimality*. The former one derives a scalar objective and uses standard single-objective optimization techniques (7; 8; 9; 10; 11). The later method is based on the concept of *Pareto dominance* and considers the set of all non-inferior solutions (12; 13). Multiple gradient methods that leverage first-order, necessary conditions for Pareto optimality have also been developed (14; 15; 16).

Similarly to MOO, existing work in MORL can be roughly divided into two main categories: *Single-policy* methods aim to maximize a single, scalarized reward. These methods essentially transform the problem into a single-objective MDP and differ mostly in the way they determine and express the preferences. Scalarization is usually performed using a weighted sum of the reward vector (17; 18; 19) or, less commonly, using linear mappings (20). Different single-policy methods are based on thresholds or lexicographical ordering (21) or different classes of preferences (22; 23). More recently, a scalarized Q-learning algorithm has been developed in (24) which uses the concept of

*corner weight* to infer and optimize over preferences. It was extended to handle dynamic preferences in (25) and to utilize the convex envelope of the Pareto front in (26). Finally, a scale-invariant supervised learning approach for encoding the preferences was developed in (27).

On the other hand, *multiple-policy* methods use the vector-valued reward signal and learn a set of policies that approximate the Pareto front. The Pareto optimal solutions are the subset of non-inferior and equally good alternative polices among all distributions in the policy space and multiple-policy methods mainly differ on the approximation scheme for the Pareto front. One common approach is to perform multiple runs (18; 17) of a single scalarized reward function over a set of preferences. Unfortunately, such methods lack scalability to high-dimensional rewards. Other approaches leverage convex approximations of the Pareto front (28) or linear approximations of the value function (29; 30) to learn optimal deterministic policies. Multi-objective fitted Q-iteration (31; 32) enables us to learn policies for all linear preferences by encapsulating the preference vector as an input to the Q-function approximator. A policy gradient method based on discrete approximations of the Pareto front was proposed in (33) and enhanced to utilize continuous approximations in (34). A gradient-based method that learns a manifold on the policy parameter space leveraging episodic exploration and importance sampling was developed in (35). Finally, a prediction-guided evolutionary algorithm which is able to find dense approximation to the Pareto front was proposed in (36).

Single-policy methods have the advantage of learning a single policy and being simple to implement. However, they suffer from instability issues, as a small change on the preferences may lead to performance collapse (37), and also rely on heuristics to infer the preferences when they are not known. On the other hand, multiple-policy methods enjoy the advantage of being able to adapt to changing preferences because the solution (being an approximation to the Pareto front) encapsulates all trade-offs between the objectives. However, this benefit comes at a higher computational cost as we typically have to learn and store multiple policies. In this paper, we aim to bridge the gab between single and multiple policy MORL methods: Our method learns a single policy along with the underlying preference vectors and is able to adapt the inferred preference to any preference distribution. Our preference vectors are inferred, not by using heuristics, but by approximating the Pareto front via a first-order condition (see Fig. 1 for an example). Our method is inspired by the multiple gradient descent algorithm for MOO introduced in (15) as well as its application in the field of multi-task supervised learning presented in (38).

**Contributions**: We propose a policy gradient method for multi-objective reinforcement learning under unknown, linear preferences. Initially, we present Pareto stationarity as a necessary, first-order condition for Pareto optimality and develop a multi-objective policy gradient algorithm which uses a projected gradient descent solver to search for and take steps in a common ascent direction for all objectives. Following that, we tackle the problem of adapting the policy gradient to any given preference distribution. We utilize our method from the first part and introduce the Pareto Policy Adaptation (PPA), a loss function that penalizes deviations between the given preference distribution and the recovered preferences.

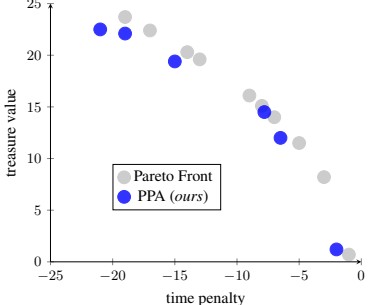

Figure 1: True and identified Pareto front for Deep Sea Treasure enviroment.

Using implicit differentiation, we are able to back-propagate the gradient of PPA bypassing the operations of the projected gradient descent solver, which makes our method applicable to real-world problems. We evaluate our method in a series reinforcement learning tasks.

The most closely related work to ours is presented in (33). The authors discuss the existence of common ascent directions, introduce a quadratic optimization problem for their computation and give two gradient-based learning algorithms. Even though our method identifies the common ascent directions using a similar optimization problem, we leverage these directions to define the PPA loss function, which is a simple and efficient way 1) to incorporate the ascent directions into any learning algorithm and 2) to account for a given preference distribution. Leveraging implicit differentiation, PPA enables us to incorporate multiple rewards and preference distributions into any modern policy gradient algorithm by a mere modification of the loss function and by implementing a projected gradient descent optimizer.

## 2 PRELIMINARIES

We consider the framework of Multi-Objective Markov Decision Process (MOMDP) $\mathcal{M} := (\mathcal{S}, \mathcal{A}, \mathcal{R}, \mathcal{P}, \gamma, \Omega, f_\Omega)$, where $\mathcal{S}$ and $\mathcal{A}$ denote the (discrete) state and action spaces, $\mathcal{P} : \mathcal{S} \times \mathcal{A} \times \mathcal{S} \to [0,1]$ is the transition kernel representing the environment dynamics, $r(s,a) \equiv \mathcal{R} : \mathcal{S} \times \mathcal{A} \to \mathbb{R}^M$ is a function mapping state-action tuples to reward vectors, $\Omega$ is the space of preferences and $f_\omega : \mathbb{R}^M \to \mathbb{R}$ a preference function which produces a scalar *utility* using a preference $\omega \in \Omega$. To declutter notation, we assume that there exists a given initial state. We consider linear preferences $f_\omega(r(s,a)) = \omega^T r(s,a)$. A fixed preference vector $\omega \in \Omega$ allows to directly compare the vectorized reward and value function by comparing their scalarized utilities, and, therefore, the MOMDP collapses to a standard MDP. A stationary control policy $\pi : \mathcal{S} \times \mathcal{A} \to \mathcal{P}(\mathcal{A})$, where $\mathcal{P}(\mathcal{A})$ is the Borel set of $\mathcal{A}$, is a function that assigns a probability distribution over actions for all states. The *value* $v^\pi : \mathcal{S} \to \mathbb{R}^{M \times |\mathcal{S}|}$ of a policy $\pi$ is defined as $v^\pi(s) := \mathbb{E}_\pi[\sum_{t=0}^\infty \gamma^t r(s_t, a_t) \mid s_0 = s]$, where $\gamma \in (0,1)$ is a discount factor and the expectation is taken under the distribution over trajectories $\tau = (s_0, a_0, s_1, a, s_2, \dots)$ obtained by starting from $s$ and following policy $\pi$ thereafter. The *discounted state-action occupation measure* under policy $\pi$ is defined as $d^\pi(s,a) := \sum_{t=0}^\infty \gamma^t P(s_t = s, a_t = a \mid \pi)$ for all states $s \in \mathcal{S}$ and allows us to write the value function compactly as $v^\pi(s) = \mathbb{E}_{d^\pi}[r(s,a) \mid s_0 = s]$.

**Definition 1.** *Considering all possible returns $\tilde{r} := \sum_{t=0}^\infty \gamma^t r(s_t, a_t)$, we define the Reward Space Pareto Coverage Set (RS-PCS) or Pareto frontier as $\mathcal{F}_r^* := \{\tilde{r} : \tilde{r} \in \mathbb{R}^M, \tilde{r}' \succ_P \tilde{r}\}$, where the symbol $\succ_P$ denotes Pareto dominance: greater or equal in all objectives and strictly greater in at least one objective. For all possible preferences in $\Omega$ and under the linear preference assumption, we define the Reward Space Convex Coverage Set (RS-CCS) as*

$$\mathcal{C}_{\mathcal{R},\Omega}^* := \{\tilde{r} : \tilde{r} \in \mathbb{R}, \exists \omega \in \Omega \text{ s.t. } \omega^T \tilde{r} \geq \omega^T \tilde{r}', \forall \tilde{r}' \in \mathbb{R}^M\}. \tag{1}$$

The set $\mathcal{C}_{\mathcal{R},\Omega}^*$ contains all discounted reward vectors that could be optimal for some preference vector in $\Omega$. Anything in $\mathcal{F}_r^*$ but not in $\mathcal{C}_{\mathcal{R},\Omega}^*$ cannot be useful under linear preferences. In that case, it suffices to restrict our analysis to $\mathcal{C}_{\mathcal{R},\Omega}^*$. Also, we assume, without loss of generality, that $\sum_m \omega_m = 1, \omega_m \geq 0, m = 1, 2 \dots M$ for all $\omega \in \Omega$, i.e., the preference vector is a convex combination of the objectives. We extend the definition of Pareto optimality from the reward space to the policy space.

**Definition 2.** *A policy $\pi$ dominates a policy $\pi'$ if and only if $v^\pi \succ_P v^{\pi'}$. A policy $\pi^*$ is called Pareto optimal if there exists no other policy $\pi$ that dominates $\pi^*$. The set $\Pi_P^*$ of all Pareto optimal policies is called Pareto policy set and its image $\mathcal{F}_\pi^* := \{v^\pi\}_{\pi \in \Pi_P^*}$ is called Pareto policy frontier. The set*

$$\mathcal{C}_{\Pi,\Omega}^* := \{\pi : \pi \in \Pi, \exists \omega \in \Omega \text{ s.t. } \omega^T v^\pi \geq \omega^T v^{\pi'}, \forall \pi' \in \Pi, \sum_m \omega_m = 1, \omega_m \geq 0\} \tag{2}$$

*is called the Policy Space Convex Coverage Set (PS-CCS).*

**Lemma 1.** *Let $\pi \in \mathcal{C}_{\Pi,\Omega}^*$ be a policy. Then, we have $\mathbb{E}_\pi[\tilde{r}] \in \mathcal{C}_{\mathcal{R},\Omega}^*$, i.e., the expected return vector under policy $\pi$ belongs in the RS-CCS.*

Lemma 1 (proof given in the Appendix) connects the PS-CCS with the RS-CCS and allows us to use the former one in place of the later. This is very convenient because searching for Pareto optimal vectors in the reward space is not practically efficient (e.g., due to the sparsity of reward signal) or useful (e.g., because they cannot be easily mapped to values or actions). By finding a policy $\pi \in \mathcal{C}_{\Pi,\Omega}^*$, Lemma 1 guarantees that the expected return vector under $\pi$ belongs in $\mathcal{C}_{\mathcal{R},\Omega}^*$ and, therefore, maps the problem of reward space exploration to policy space exploration. Additionally, Lemma 1 imposes no restrictions on the class of policies and allows us to adopt the common practice of parameterizing the policy by high-dimensional vector $\theta \in \Theta$ (e.g., a neural network). The definition of PS-CCS given Eq. equation 3 carries over to this parametric class as is, i.e., we can define the PS-CCS in the policy parameter space as follows

$$\mathcal{C}_{\Theta,\Omega}^* := \{\theta : \theta \in \Theta, \exists \omega \in \Omega \text{ s.t. } \omega^T v^{\pi_\theta} \geq \omega^T v^{\pi_{\theta'}}, \forall \theta' \in \Theta\}. \tag{3}$$

## 3 LEARNING WITH UNKNOWN PREFERENCES

In this section, we propose a new policy gradient algorithm for MORL under unknown preferences. The key idea of our approach is to infer the preference vector $\omega$, use that $\omega$ as scalarization coeffi-

cient for the multi-objective criterion and ascent on the gradient of the scalarized value. To search for a Pareto optimal policy, we leverage a fist-order condition known as Pareto stationarity.

**Definition 3** (Pareto stationarity). *A policy $\pi_\theta$ is called Pareto stationary if and only if $\sum_{m=1}^{M} \omega_m \nabla_\theta \mathbb{E}_{d^{\pi_\theta}}[r_m(s,a)] = 0$ for some $\omega_m \in [0,1]$, $m = 1, 2 \ldots M$.*

Pareto stationarity implies that there exists a vanishing convex combination of the objectives' gradients and is a necessary condition for Pareto optimality (15). The following theorem connect the PS-CCS with Pareto stationarity.

**Theorem 1.** *Let $\pi_\theta \in \mathcal{C}^*_{\Theta,\Omega}$ be a policy in the PS-CCS. Then, $\pi_\theta$ is Pareto stationary.*

*Proof.* Let $\pi_\theta \in \mathcal{C}^*_{\Theta,\Omega}$ be a policy and $\boldsymbol{\omega}$ the corresponding preferences, i.e., we have $\boldsymbol{\omega}^T \boldsymbol{v}^{\pi_\theta} \geq \boldsymbol{\omega}^T \boldsymbol{v}^{\pi_{\theta'}}$, $\forall \theta' \in \Theta$. The last relation can be written as $\boldsymbol{\omega}^T \mathbb{E}_{\pi_\theta}[\tilde{\boldsymbol{r}}] \geq \boldsymbol{\omega}^T \mathbb{E}_{\pi_{\theta'}}[\tilde{\boldsymbol{r}}]$, $\forall \theta' \in \Theta$. This implies that $\theta$ is a global maximizer of $\boldsymbol{\omega}^T \mathbb{E}_{\pi_\theta}[\tilde{\boldsymbol{r}}]$ or, equivalently, that its gradient at $\theta$ vanishes. Therefore, we have

$$\nabla_\theta \boldsymbol{\omega}^T \mathbb{E}_{d^{\pi_\theta}}[\tilde{\boldsymbol{r}}] = \sum_{m=1}^{M} \omega_m \nabla_\theta \mathbb{E}_{d^{\pi_\theta}}[r_m(s,a)] = 0, \quad (4)$$

which corresponds to the necessary condition for Pareto stationarity as per Def. 3. $\qquad\square$

Theorem 1 essentially states that Pareto stationarity is a necessary condition for a policy to belong to PS-CCS. Additionally, as stated in the proof, the convex combination of the objectives that leads to vanishing gradients coincides with the preferences that make the underlying policy part of the PS-CCS. This presents us with a strong necessary condition for recovering the unknown preferences and for identifying an optimal policy. This is because the problem of finding a Pareto stationary is tractable via gradient methods. In fact, if a policy is not Pareto stationary then there exists a common ascent direction for all objectives. To find this descent direction, we use the following theorem.

**Theorem 2** ((15), Theorem 1). *Consider the following quadratic optimization problem*

$$\min_{\omega_1, \ldots \omega_M} \left\{ \left\| \sum_{m=1}^{M} \omega_m \nabla_\theta \mathbb{E}_{d^{\pi_\theta}}[r_m(s,a)] \right\|_2^2 \; s.t \; \sum_{m=1}^{M} \omega_m = 1, \; \omega_m \geq 0, \; \forall m \right\}. \quad (P_A)$$

*Let $\boldsymbol{\omega}^*$ be the optimal solution of Problem $P_A$ and $\mu^*$ the underlying minimal-norm objective. Then:*

1. *either $\mu^* = 0$ and $\pi_{\theta^*}$ is a Pareto stationary policy*

2. *or $\sum_{m=1}^{M} \omega_m^* \nabla_\theta \mathbb{E}_{d^{\pi_\theta}}[r_m(s,a)]$ is a common ascent direction for all objectives.*

Problem $P_A$ is essentially equivalent to the formulation of the famous *Minimum-Norm Point* problem, which arises many fields such as optimal control (39), submodular minimization (40) and portfolio optimization (41). The MNP problem has been well studied and several combinatorial and recursive algorithms have been proposed (42; 43; 44). Such methods search for the *exact* solution and typically have exponential run-time (45), which prevents their application in high-dimensional control policies. Therefore, we adopt convex optimization approach for Problem 2. We start by noting that the constrain set $\mathcal{E} = \{\boldsymbol{\omega} \in \mathbb{R}^M : \sum_{m=1}^{M} \omega_m = 1, \; \omega_m \geq 0, \; \forall m = 1, 2 \ldots M\}$ is essentially the unit simplex and the projection problem, i.e., $\Pi_\mathcal{E}(\boldsymbol{\omega}) = \arg\min_{\boldsymbol{\delta} \in \mathcal{E}} \|\boldsymbol{\omega} - \boldsymbol{\delta}\|$, can be efficiently solved. In fact, there exists a unique dual variable $\tau \in \mathbb{R}$ such that

$$\Pi_\mathcal{E}(\boldsymbol{\omega}) = (\boldsymbol{\omega} - \tau)_+, \quad (5)$$

where $(x)_+ = \max(x, 0)$. To find the dual variable, we enforce the equality constraint, i.e., $\sum_m (\omega_m - \tau)_+ = 1$. This non-linear equation of a single variable can be solved very efficiently using the Newton method. This suggests a projected gradient descent solver for solving Problem $P_A$ and recovering the unknown preferences.

Our method (Algorithm 1) extends the classical REINFORCE algorithm (46) to take gradient steps in the direction given by the weighted combination of the multi-objective gradient, with the weights being the preferences recovered by the projected gradient descent solver. Lemma 1, Theorem 1 and 2 essentially state that Algorithm 1 learns arbitrary policies in the Convex Coverage Set along with

their underlying preference vectors and provide theoretical justification for our method. Algorithm 1 is simple to implement, compatible with all policy gradient and actor-critic methods and provides a practical algorithm for learning Pareto optimal policies under unknown preferences. However, it suffers from a subtle drawback: we have no control over the identified preferences and they cannot be optimized to account for any given preference distribution. This is the problem of policy adaptation which we address in the next section.

It is worth comparing our projected gradient descent solver with (38), where the authors used the Frank-Wolfe method for Problem $P_A$, albeit in the context of supervised learning. Both algorithms have the same convergence rate of $\mathcal{O}(1/N)$ and since the projection to the constraint set can be easily calculated, we opted for the projected gradient descent optimizer. The overhead added by solving for the dual variable $\tau$ is minimal because the Newton's method is known to enjoy quadratic convergence rate and is very efficient in practice. The trade-off of using Frank-Wolfe is that requires to obtain the feasible minimizer of the linear approximation of the objective around the current iterant, which is typically solved approximately and worsens the convergence rate by a multiplicative constant (47). Nonetheless, we expect that both methods can be used interchangeably in practice.

---

**Algorithm 1:** Multi-Objective Policy Gradient

**Inputs:** Initial parameters $\theta_0$, learning rate $\alpha$

1 **foreach** $k = 0, 1, \ldots$ **do**
2     Collect trajectories $\mathcal{D} = \{\tau_i\}$, $\tau_i \sim \pi_{\theta_k}$
3     Compute rewards-to-go $\boldsymbol{r}_t$
4     $g_m \leftarrow \frac{1}{|\mathcal{D}|} \sum_{\tau,t} \boldsymbol{r}_t \nabla_\theta \log \pi_{\theta_k}(a_t|s_t)$
5     $(\omega_1, \ldots \omega_M) \longleftarrow \texttt{PGDSolver}(g_1, \ldots g_M, \alpha)$
6     $\theta_{k+1} \leftarrow \theta_k + \sum_{m=1}^{M} \omega_m g_m$
7 **end**
8
9 **procedure** $\texttt{PGDSolver}(g_1, \ldots, g_M, \alpha)$:
10     Initialize feasible $\boldsymbol{\omega} = (\omega_1, \ldots \omega_M)$
11     **while** *not converged* **do**
12        Find $\tau$ solving $\sum_m (\omega_m - \tau)_+ = 1$
13        $\omega_m \leftarrow \omega_m + \alpha g_m^T \sum_n \boldsymbol{e}_n g_n$, $m = 1 \ldots M$
14        $\boldsymbol{\omega} \leftarrow (\boldsymbol{\omega} - \tau)_+$
15     **end**
16     **return** $(\omega_1, \ldots \omega_M)$

---

## 4 PARETO POLICY ADAPTATION

The policy gradient algorithm presented in the previous section finds policies that are optimal (i.e., Pareto stationary) for the inferred preferences. In this section, we show how to address the problem of adapting the policy to be optimal for a given distribution $\mathcal{D}_{\boldsymbol{\omega}}$ over preferences. We start by expressing the optimal solution of Problem $P_A$ as a function of the policy parameters $\boldsymbol{\omega}^* := \boldsymbol{\omega}^*(\theta) = (\omega_1(\theta), \omega_2(\theta), \ldots \omega_M(\theta))$. We introduce the *Pareto Policy Adaptation* loss function as follows:

$$\mathcal{L}(\theta) = \mathbb{E}_{d^{\pi_\theta}, \boldsymbol{\omega}} \left[ \lambda \sum_{m=1}^{M} \omega_m^*(\theta) r_m(s, a) - (1-\lambda) \|\boldsymbol{\omega}^*(\theta) - \boldsymbol{\omega}\|_2^2 \right], \quad (6)$$

where $\lambda \in [0, 1]$ is a weight that trades off between the two losses. The first term of PPA is the scalarized reward, with the scalarization coefficient being the inferred preferences. The second term penalizes the expected deviation between the given and inferred preferences. To further motivate the PPA loss, consider the special case of a single, given preference vector $\boldsymbol{\omega}^0$. In this case, the MOMDP collapses to a standard MDP, with $\boldsymbol{\omega}^0$ being the scalarization coefficient for the reward and value functions. Additionally, assume that the policy $\pi_\theta$ has been *adapted* to that preference vector, i.e., it achieves $\boldsymbol{\omega}^*(\theta) = \boldsymbol{\omega}^0$ for some parameter vector $\theta$. Then, Eq. 6 reduces to $\mathcal{L}(\theta) = \mathbb{E}_{d^{\pi_\theta}} \left[ \sum_{m=1}^{M} \omega_m^0 r_m(s, a) \right]$. Maximization of this loss corresponds to taking ascent steps along $\sum_m \omega_m^0 \nabla_\theta \mathbb{E}_{d^{\pi_\theta}} [r_m(s, a)]$, which, per Theorem 2, is an ascent direction guaranteed to increase all objectives. This simplified scenario essentially states the if we *exactly* match the preferences learned from Problem $P_A$ to the given preference vector $\boldsymbol{\omega}^0$, then not only the PPA loss function reduces to the loss function for the scalarized reward, but also the ascent steps along its gradient lead to concurrent improvements of all objectives. We show a geometric illustration of the PPA loss function in Fig. 2.

Despite its simplicity, one important obstacle in using the PPA loss to train agents in practice is that the optimal solution to the Problem $P_A$ is analytically known only for $M = 2$ (solution is presented

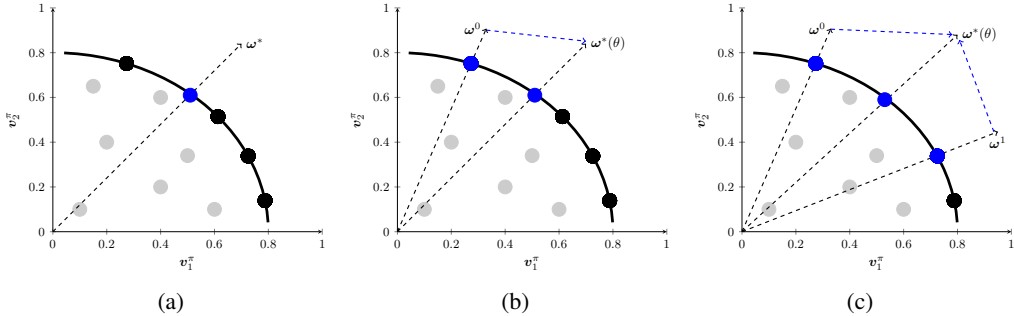

Figure 2: (a) The Convex Coverage Set (CCS) for a 2-dimensional value function. Algorithm 1 identifies arbitrary solutions on the Pareto front. (b) Illustration of the PPA loss function for a single preference vector $\boldsymbol{\omega}^0$: by minimizing the distance $\left\|\boldsymbol{\omega}^*(\theta) - \boldsymbol{\omega}^0\right\|$, the solution converges to the optimal solution corresponding to preference $\boldsymbol{\omega}^0$. (c) PPA loss function for two preference vectors $\boldsymbol{\omega}^0, \boldsymbol{\omega}^1$: the minimizer of $\left\|\boldsymbol{\omega}^*(\theta) - \boldsymbol{\omega}^0\right\| + \left\|\boldsymbol{\omega}^*(\theta) - \boldsymbol{\omega}^1\right\|$ (also called the *geometric median*) gives the optimal solution because it maximizes the projections onto the preference vectors.

in the Appendix). For $M > 2$, it can rather only be approximated by the iterative projected gradient descent solver introduced in Algorithm 1. This makes the backward pass challenging because the gradients need to be back-propagated through the (perhaps thousands) iterations of the projected gradient descent solver. Since back-propagation is known to be a very expensive operation compared to the forward pass, this could make our approach inefficient for practical problems. The problem of back-propagating through solutions of numerical solvers arises in several deep learning architectures such as deep equilibrium models (48) and optimization-based neural network layers (49). Luckily, the implicit value theorem allows us to back-propagate the gradient bypassing the operations of the projected gradient descent solver.

**Theorem 3.** *Let $l := l(\boldsymbol{\omega}^*(\theta))$ be a generic loss function that depends explicitly on the the optimal solution $\boldsymbol{\omega}^*(\theta)$ of Problem $P_A$. Then, the gradient of $l$ with respect to the policy parameters is*

$$\nabla_\theta l = \begin{bmatrix} H^T(\theta) & \mathbf{0} & \mathbf{0} \end{bmatrix} \begin{bmatrix} G(\theta) & -I_M & \boldsymbol{e} \\ -D(\boldsymbol{\mu}^*(\theta)) & -D(\boldsymbol{\omega}^*(\theta)) & \mathbf{0} \\ \boldsymbol{e}^T & \mathbf{0} & \mathbf{0} \end{bmatrix}^{-T} \begin{bmatrix} \frac{dl}{d\boldsymbol{\omega}^*} \\ \mathbf{0} \\ \mathbf{0} \end{bmatrix}, \qquad (7)$$

*where $[G(\theta)]_{m,n} = 2g_m^T(\theta)g_n(\theta)$, $[H(\theta)]_{m,\cdot} = -\sum_n \omega_n^*(\theta)\nabla_\theta^T(g_n^T(\theta)g_m(\theta))$, $g_m(\theta) = \nabla_\theta \mathbb{E}_{d^{\pi_\theta}}[r_m(s, a)]$, $\boldsymbol{\mu}^*(\theta)$ is the optimal dual variable, $\boldsymbol{e}$ and $\mathbf{0}$ are the identity and zero vectors of appropriate dimensions, $I_M$ is the identity matrix of dimension $M$ and $D(\cdot)$ create a diagonal matrix from a vector. When applied on a vector quantity, the operator $\nabla_\theta$ denotes the Jacobian matrix and the operator $\nabla_\theta^T$ denotes the transpose of the underlying gradient or Jacobian.*

The proof of Theorem 3 (given in the Appendix) leverages the implicit value theorem which enables us to differentiate the KKT optimality conditions of Problem $P_A$ with respect to the policy parameters. Observe that matrix $H(\theta)$ is essentially of dimension $M \times d_\theta$, $d_\theta$ being the dimension of $\theta$. This dependence on $d_\theta$ makes the solution of eq:nmatrix inefficient, as we need to compute and store a matrix of size $(2M + 1)d_\theta$. However, in practice, we are not interested in knowing the full gradient of $\boldsymbol{\omega}^*(\theta)$, but rather in back-propagating the gradient of the loss function; Eq. 7 has an appealing form which allows us to use vector-Jacobian products (VJPs) to evaluate the gradient $\nabla_\theta l$ one row at a time. This frees us from having to store matrix $H(\theta)$ and compute its product with another matrix. We observe that the solution in Eq. 7 requires knowledge of the optimal dual variable $\boldsymbol{\mu}^*(\theta)$, which is not directly available. However, the optimal primal solution $\boldsymbol{\omega}^*(\theta)$ given by the projected gradient descent solver (Algorithm 1) can be utilized to obtain $\boldsymbol{\mu}^*(\theta)$ by substituting it into the KKT optimality conditions of Problem $P_A$ (details in the Appendix) and solving the resulting linear system. That system has $M + 1$ variables, therefore its solution adds minimal computational overhead.

It is worth discussing similarities and differences between PPA and EQL (26). PPA retrieves and adapts to preferences during the training phase, whereas EQL first learns a preference-dependant policy and then retrieves preferences using a combination of stochastic search and policy gradient.

Also, the loss functions are partially similar: the penalty term in PPA is the MSE loss between the given and inferred preferences, whereas the EQL loss function is the MSE loss of vector Q-values. Finally, the optimal policy obtained under a given preference distribution is not, in general, optimal under a different distribution and the model has to be fine-tuned. We expect that we will not need to re-train the network entirely and that we can leverage the previous policy to find the new one. This is an interesting direction for future work.

## 5 EXPERIMENTAL EVALUATION

In this section, we evaluate the performance of our proposed method. In our implementation, we extended the popular Proximal Policy Optimization (PPO) (50) algorithm so that it can handle our PPA loss function. We implemented all experiments on PyTorch using modified versions of OpenAI Gym environments. We give the details of our experimental setup in the Appendix.

**Domains**: We evaluate on 4 environments (details given in the Appendix): (a) Multi-Objective GridWorld (MOWG): a variant of the classical gridworld, (b) Deep Sea Treasure (DST): a slightly modified version of classical multi-objective reinforcement learning enviroment (51), (c) Mulit-Objective SuperMario (MOSM): modified, multi-objective variant of the popular video game that has a 5-dimensional reward signal and (d) Multi-Objective MuJoCo (MOMU): a modified version of the MuJoCo physics simulator, focusing on locomotion tasks.

**Ablation Study and Baselines**: To highlight the extent to which our results are driven by the PPA loss function, we perform an ablation study. In more details, we consider the following three agents: (a) Pareto Policy Adaptation (PPA): This is our original loss function as introduced in Sec. 4. (b) non-adaptive PPA (na-PPA): Same as PPA but the preferences are static, i.e., the policy is not adapted to the preference distribution. This agent is essentially Algorithm 1 without the PPA loss function. (c) Fixed Preferences: Same as na-PPA but the preferences are given. This corresponds to optimizing the scalarized objective. Additionally, we compare the performance of our method against a range of state-of-the-art baselines. In particular, we compare against: the Pareto Following Algorithm (PFA) (33), the Conditional Neural network with Optimistic Linear Support (CN-OLS)(24; 25), the Conditioned Network with Diverse Experience Replay (CN-DER) (25), the Envelope Q-Learning (EQL) (26) and the Prediction-Guided Multi-Objective Reinforcement Learning (PGMORL) (36).

**Evaluation Metrics**: To evaluate the performance of our method and compare against the baselines, we use the following metrics:

1. *HyperVolume* (HV): Let $\mathcal{F}$ be a reward space Pareto front approximation and $r_0$ be a reference vector. Then, we define the *HyperVolume* as $\text{HV} := \int \mathbb{1}_{H(P)}(z)dz$, where $H(P) = \{z \in \mathbb{R}^M : \exists r \in \mathcal{F} : r_0 \prec_P z \prec_P r\}$ and $\mathbb{1}(\cdot)$ is the indicator function.

2. *Pareto Dominance* (PD): Let $r_t^1$, $r_t^2$ be the reward vectors of trajectories sampled from agent 1 and 2, respectively. We define the *Pareto Dominance* metric as $\text{PD}_{1,2} := \frac{1}{T}\sum_{t=0}^T \mathbb{1}(r_t^1 \succ_P r_t^2)$, where $\mathbb{1}(\cdot)$ is the indicator function. This is essentially the fraction of time when the reward vector under agent 1 Pareto dominates the reward vector under agent 2. Note that $\text{PD}_{1,2} + \text{PD}_{2,1} \leq 1$.

3. *Utility* (UT): For a reward vector $r_t$ sampled from any agent, we define the *Utility* metric as $\text{UT} := \mathbb{E}_{\boldsymbol{\omega}}[\frac{1}{T}\sum_{t=0}^T \boldsymbol{\omega}^T r_t]$, where the expected value is evaluated by sampling uniformly from the preference space $\Omega$ or from the preference distribution $\mathcal{D}_{\boldsymbol{\omega}}$.

**Main Results**: Figure 3 shows the results of the ablation study conducted on the MOGW and DST domains. We compare against the non-adaptive and the fixed-preferences agents. For the later case, we pick three different preference vectors: $(\omega_1, \omega_2) = (0.5, 0.5)$, $(\omega_1, \omega_2) = (0.75, 0.25)$ and $(\omega_1, \omega_2) = (0.25, 0.75)$. These plots facilitate the visual identification of Pareto dominant agents. Focusing on the MOGW (upper row), we observe that our PPA agent converges to a policy that yields a higher reward and lower time-penalty than any other agent, i..e, PPA Pareto dominates all other agents. While a similar statement cannot be made for the DST domain, we note that PPA is not Pareto dominated by any other agent and the only agent that it does not dominate is the fixed-preference one for $(\omega_1, \omega_2) = (0.5, 0.5)$. On the other hand, the non-adaptive PPA (na-PPA) seems to perform relatively poor. However, there exists some merit in using it because, in contrast to the

fixed-preference agents, `na-PPA` does not rely on being given a preference vector. It is also worth noting that in the DST domain the preference vector $(\omega_1, \omega_2) = (0.25, 0.75)$ gives very poor results, which implies that handpicking the preferences might lead to performance collapse. Overall, these results suggest that `PPA` steadily achieves better performance and is superior than the other agents. It is interesting to observe that the reward and time penalty curves are quite similar in the MOWG domain. This agrees with our intuition because in the MOGW domain the reward and time penalty have an "inverse" relationship (i.e., the sooner we reach the bottom right, the higher the reward).

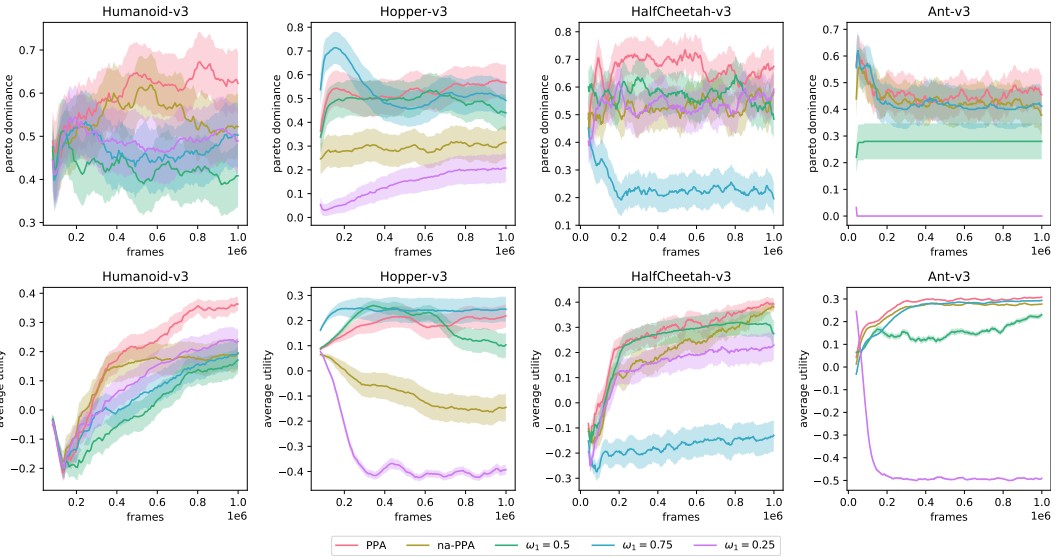

Figure 4: Learning curves on MuJoCo environments. We show the Pareto Dominance (`PD`) and Average Utility (`UT`) metrics. Experiments are run for 10 random seeds. We compare against the non-adaptive PPA and the fixed-preferences agents.

|  |  | PFA | CN-OLS | CD-DER | EQL | PGMORL | nA-PPA | PPA |
|---|---|---|---|---|---|---|---|---|
| **MOGW** | HV | 2.52 | 2.36 | 2.65 | 3.10 | 3.15 | 2.35 | **3.32** |
|  | PD | 0.52 | 0.51 | 0.46 | 0.53 | 0.55 | 0.33 | **0.65** |
|  | UT | 0.23 | 0.19 | $-0.12$ | 0.18 | 0.28 | 0.08 | **0.28** |
| **DST** | HV | 36.32 | 35.36 | 35.65 | 35.12 | 39.35 | 34.20 | **40.12** |
|  | PD | 0.46 | 0.55 | 0.39 | 0.48 | 0.52 | 0.38 | **0.75** |
|  | UT | 0.19 | 0.14 | $-0.11$ | 0.20 | **0.24** | 0.12 | 0.21 |
| **MOSM** | HV | - | 1146.14 | 1364.14 | 1134.12 | 1401.14 | 1125.18 | **1442.14** |
|  | PD | - | 0.47 | 0.41 | 0.50 | 0.59 | 0.42 | **0.64** |
|  | UT | - | 0.15 | $-0.07$ | 0.24 | 0.26 | 0.14 | **0.27** |
| **Humanoid-v3** | HV | - | 4.39 | 5.18 | 4.95 | 4.65 | 4.02 | **5.15** |
|  | PD | - | 0.41 | 0.46 | 0.54 | 0.52 | 0.33 | **0.55** |
|  | UT | - | 0.19 | 0.22 | 0.41 | 0.30 | 0.28 | **0.48** |
| **Hopper-v3** | HV | - | 3.18 | 2.98 | 3.44 | **3.74** | 2.23 | 3.73 |
|  | PD | - | 0.43 | 0.49 | 0.45 | 0.52 | 0.45 | **0.70** |
|  | UT | - | $-0.23$ | 0.14 | 0.22 | 0.21 | 0.12 | **0.25** |
| **HalfCheetah-v3** | HV | - | 5.17 | 5.38 | 5.39 | **5.77** | 4.18 | 5.68 |
|  | PD | - | 0.37 | 0.38 | 0.51 | 0.55 | 0.45 | **0.62** |
|  | UT | - | $-0.10$ | 0.11 | 0.22 | 0.20 | 0.18 | **0.31** |
| **Ant-v3** | HV | - | 5.95 | 5.47 | 6.30 | 6.35 | 5.25 | **6.45** |
|  | PD | - | 0.26 | 0.30 | 0.44 | 0.41 | 0.32 | **0.52** |
|  | UT | - | 0.36 | 0.31 | 0.34 | **0.55** | 0.39 | **0.55** |

Table 1: Comparison of MORL agents using the HyperVolume (`HV`), Pareto Dominance (`PD`), and Average Utility (`UT`) metrics. Possible values for each metric lie in the range $[0, \infty]$, $[0, 1]$, and $[-1, 1]$, respectively. In all cases, higher values indicate better performance. We evaluate the agents using 10 random seeds.

In the next line of experiments, we evaluate the performance of `PPA` in continuous control tasks. We modified the MuJoCo Gym interface to output 2-dimensional rewards on each step. The first

one is the commonly used reward in such environments, i.e., the difference between two sequential positions of the robot. The second one is the energy consumed by the robot. We show the results on Fig. 4. We compare against the ablated agents and show the Pareto Dominance (PD) and Average Utility (UT) metrics. The only case where our method does not attain the best performance is the Hopper-v3 environment under the UT metric, where the performance of the fixed-preferences agent $(\omega_1, \omega_2) = (0.75, 0.25)$ slightly surpasses the performance of PPA. However, observe that the same agent attains very poor performance in the HalfCheetah-v3 enviroment. This suggests that, given the complex inter-dependence between the two rewards, simply hand-crafting a preference vector is far from optimal and a preference vector that yields good performance in one environment may perform poorly in a different environment or judged under a different metric.

In the last line of experiments, we compare our method against state-of-the-art MORL algorithms (Table 1). Since PD is a pair-wise metric, we first compute the pair-wise $\mathrm{PD}_{i,j}$ values for all agents and then average-out the $j$ component to obtain a single value for each agent. Note that this metric is specific to the ensemble of agents that we consider and needs to be re-calculated if we compare a different set of agents. The UT metric is calculated by sampling preferences from the preference space of each domain. We assume assume a uniform preference distribution and sample 1000 points for the 2-dimensional preference case (MOGW, DST, MUMO) and 100 across each dimension for the MOSM case, where the preference space has 4 dimensions. Additionally, we scale the re-

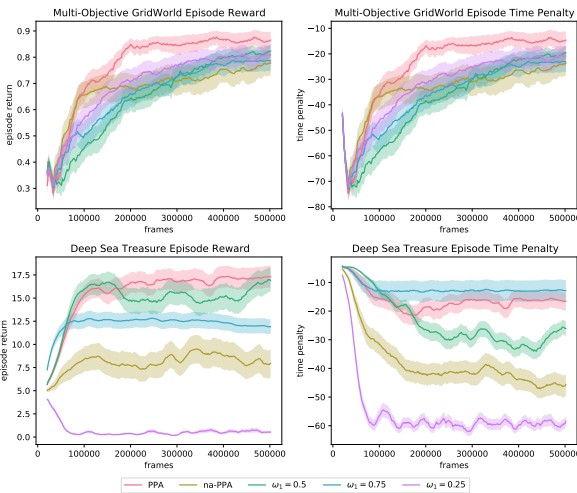

Figure 3: Results on the MOGW and DST environments.

wards in the range $[-1, 1]$ across each dimension. This scaling is important when calculating the average utility but has no particular influence on the Pareto dominance or hypervolume. Observe that some agents have negative utility, which is indeed possible given the aforementioned reward scaling. Our method systematically outperforms all other baselines across all domains, which is another strong empirical finding for its performance.

## 6 CONCLUSION

We introduced a policy gradient algorithm for Multi-Objective Reinforcement Learning (MORL) under linear preferences. By approximating the Pareto front via a first-order necessary condition, our method learns a *single* policy that can be adapted to *any* preference distribution. The two pillars of our method is a projected gradient descent solver that searches for common ascent direction for all objectives and a novel *Pareto Policy Adaptation* (*PPA*) loss function that leverages the solutions of that solver to adapt the policy to any preference distribution. Our method enjoys the advantages of both single and multiple policy MORL algorithms while being simple to implement and compatible with all policy gradient and actor-critic methods. We demonstrated the effectiveness of our method in several reinforcement learning tasks.

### ACKNOWLEDGEMENTS

The authors gratefully acknowledge the support by the National Science Foundation under the Career Award CPS/CNS-1453860, the NSF award under Grant Numbers CCF-1837131, MCB-1936775, CNS-1932620, and CMMI 1936624 and the DARPA Young Faculty Award and DARPA Director's Fellowship Award, under Grant Number N66001-17-1-4044, and a Northrop Grumman grant. The views, opinions, and/or findings contained in this article are those of the authors and should not be interpreted as representing the official views or policies, either expressed or implied by the Defense Advanced Research Projects Agency, the Department of Defense or the National Science Foundation. This work was also supported by the National Science Foundation under the CAREER Award SHF-2048094, FMitF award CCF-1837131, CPS award CNS-1932620, Toyota RD, and Northrop-Grumman Aerospace Systems.

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

## A   APPENDIX

## A   THEORETICAL FRAMEWORK

In the first part of the Appendix, we give the proofs omitted in Sections 2 and 4 as well as the analytical solution of Problem $P_A$ for the special case of 2-dimensional reward signals. For the ease of reference, we re-state the underlying theorem.

### A.1   PROOFS

**Lemma 1.** *Let $\pi \in \mathcal{C}^*_{\Pi,\Omega}$ be a policy. Then, we have $\mathbb{E}_\pi[\tilde{r}] \in \mathcal{C}^*_{\mathcal{R},\Omega}$, i.e., the expected return vector under policy $\pi$ belongs in the RS-CCS.*

*Proof.* Consider a policy $\pi \in \mathcal{C}^*_{\Pi,\Omega}$ and a policy $\pi' \in \Pi$ such that $d^{\pi'}(s,a) = 0$ wherever $d^\pi(s,a) = 0$. Then, we can write

$$\boldsymbol{v}^{\pi'} = \sum_{s,a} d^{\pi'}(s,a)\boldsymbol{r}(s,a) = \sum_{s,a} d^\pi(s,a)\frac{d^{\pi'}(s,a)}{d^\pi(s,a)}\boldsymbol{r}(s,a) = \mathbb{E}_{d^\pi}[\boldsymbol{r}'(s,a)], \tag{8}$$

where we used the importance sampling identity and defined $\boldsymbol{r}'(s,a) := d^{\pi'}(s,a)/d^\pi(s,a)\boldsymbol{r}(s,a)$. Per Def. 2, there exists $\boldsymbol{\omega} \in \Omega$ such that $\boldsymbol{\omega}^T \boldsymbol{v}^\pi \geq \boldsymbol{\omega}^T \boldsymbol{v}^{\pi'}$. Leveraging the fact that the value function can be expressed as $\boldsymbol{v}^\pi = \mathbb{E}_\pi[\tilde{r}]$ as well as Eq. equation 8, we have $\boldsymbol{\omega}^T \mathbb{E}_\pi[\tilde{r}] \geq \boldsymbol{\omega}^T \mathbb{E}_\pi[\tilde{r}']$. This implies that $\mathbb{E}_\pi[\tilde{r}] \in \mathcal{C}^*_{\mathcal{R},\Omega}$ and concludes the proof. □

**Remark 1.** *Under the additional assumption of normalized rewards, we can prove the converse of Lemma 1. However, since we are are concerned with approximating $\mathcal{C}^*_{\mathcal{R},\Omega}$ by $\mathcal{C}^*_{\Pi,\Omega}$ (and not the other way around), we refrain from making that additional assumption to prove the converse of Lemma 1.*

**Theorem 3.** *Let $l := l(\boldsymbol{\omega}^*(\theta))$ be a generic loss function that depends explicitly on the the optimal solution $\boldsymbol{\omega}^*(\theta)$ of Problem $P_A$. Then, the gradient of $l$ with respect to the policy parameters is*

$$\nabla_\theta l = \begin{bmatrix} H^T(\theta) & \boldsymbol{0} & \boldsymbol{0} \end{bmatrix} \begin{bmatrix} G(\theta) & -I_M & \boldsymbol{e} \\ -D(\boldsymbol{\mu}^*(\theta)) & -D(\boldsymbol{\omega}^*(\theta)) & \boldsymbol{0} \\ \boldsymbol{e}^T & \boldsymbol{0} & \boldsymbol{0} \end{bmatrix}^{-T} \begin{bmatrix} \frac{dl}{d\boldsymbol{\omega}^*} \\ \boldsymbol{0} \\ \boldsymbol{0} \end{bmatrix}, \tag{9}$$

*where $[G(\theta)]_{m,n} = 2g_m^T(\theta)g_n(\theta)$, $[H(\theta)]_{m,\cdot} = -\sum_n \omega_n^*(\theta)\nabla_\theta^T(g_n^T(\theta)g_m(\theta))$, $\boldsymbol{e}$ and $\boldsymbol{0}$ are the identity and zero vectors of appropriate dimensions, $I_M$ is the identity matrix of dimension $M$ and $D(\cdot)$ create a diagonal matrix from a vector.*

*Proof.* We start by writing the Lagrangian of Problem $P_A$

$$L(\boldsymbol{\omega}, \lambda, \boldsymbol{\mu}) = \left\| \sum_{m=1}^M \omega_m g_m(\theta) \right\|^2 + \lambda\big( \sum_{m=1}^M \omega_m - 1 \big) + \sum_{m=1}^M \mu_m \omega_m, \tag{10}$$

where $g_m(\theta) = \nabla_\theta \mathbb{E}_{d^{\pi_\theta}}[r_m(s,a)]$. Let $(\boldsymbol{\omega}^*(\theta), \lambda^*(\theta), \boldsymbol{\mu}^*(\theta))$ be the policy-dependant optimal primal and dual variables. The KKT conditions for stationarity, primal feasibility and complementary slackness are

$$\frac{\partial L}{\partial \omega_m^*(\theta)} = 2 \sum_{n=1}^M \omega_n^*(\theta)g_n^T(\theta)g_m(\theta) + \lambda^*(\theta) + \mu_m^*(\theta) = 0, \; m = 1, 2 \ldots M \tag{11a}$$

$$\frac{\partial L}{\partial \lambda^*(\theta)} = \lambda^*(\theta)\big( \sum_{m=1}^M \omega_m^*(\theta) - 1 \big) = 0 \tag{11b}$$

$$\frac{\partial L}{\partial \mu_m^*(\theta)} = \mu_m^*(\theta)\omega^*(\theta) = 0, \; m = 1, 2 \ldots M, \tag{11c}$$

By taking the gradient of the above equations with respect to the policy parameters, we obtain

$$2 \sum_{n=1}^M \nabla_\theta \omega_n^*(\theta)g_n^T(\theta)g_m(\theta) + 2 \sum_{n=1}^M \omega_n^*(\theta)\nabla_\theta\big(g_n^T(\theta)g_m(\theta)\big) +$$
$$\nabla_\theta \lambda^*(\theta) + \nabla_\theta \mu_m^*(\theta) = 0, \; m = 1, 2 \ldots M, \tag{12a}$$

$$\sum_{m=1}^M \nabla_\theta \omega_m^*(\theta) = 0, \tag{12b}$$

$$\nabla_\theta \mu_m^*(\theta)\omega^*(\theta) + \mu_m^*(\theta)\nabla_\theta\omega^*(\theta) = 0, \; m = 1, 2 \ldots M. \tag{12c}$$

Using matrix notation, we can rewrite the above system of equations as follows:

$$\begin{bmatrix} G(\theta) & -I_M & \boldsymbol{e} \\ -D(\boldsymbol{\mu}^*(\theta)) & -D(\boldsymbol{\omega}^*(\theta)) & \mathbf{0} \\ \boldsymbol{e}^T & \mathbf{0} & \mathbf{0} \end{bmatrix} \begin{bmatrix} \nabla_\theta^T \boldsymbol{\omega}^*(\theta) \\ \nabla_\theta^T \boldsymbol{\mu}^*(\theta) \\ \nabla_\theta^T \lambda^*(\theta) \end{bmatrix} = \begin{bmatrix} H(\theta) \\ \mathbf{0} \\ \mathbf{0} \end{bmatrix}. \tag{13}$$

By inverting Eq. equation 13 and using the chain rule $\nabla_\theta l(\omega(\theta)) = \frac{\partial l}{\partial \omega} \nabla_\theta l$ we obtain the claimed result. $\qquad\square$

We observe that the solution of Eq. equation 13 requires knowledge of the optimal dual variable $\boldsymbol{\mu}^*(\theta)$, which is not directly available when we use the projected gradient descent solver of Algorithm 1; that solver only gives us the optimal primal variable $\boldsymbol{\omega}^*(\theta)$. However, we can substitute that optimal $\boldsymbol{\omega}^*(\theta)$ into the KKT conditions given by Eq. (11a–11c) and solve the resulting linear system of $M + 1$ variables to find the optimal dual variables. The number of variables in that system depends only on the dimension of the reward signal $M$ and not on the dimension of $\theta$, therefore it adds minimal computational overhead. This step would not be necessary if we chose a primal-dual method for solving Problem $P_A$. However, primal-dual methods may add to the complexity of the implementation, which may not be worthy given the simplicity of Problem $P_A$. Exploring such a trade-off is an interesting direction for future work.

## A.2 PARETO STATIONARITY: 2-DIMENSIONAL CASE

In general, the problem of finding common ascent directions (Problem $P_A$) can be solved either *exactly* using the methods from the minimum norm points literature (as discussed in Sec. 3) or *approximately* using convex optimization methods. Due to the dimensionality of the gradient vector, the former class of methods are not applicable in our case. Therefore, we resorted to the projected gradient descent solver presented in Algorithm 1. This solver is essentially an internal optimizer which we run before every parameter update. Luckily, Problem $P_A$ is convex and the gradient descent optimizer converges rapidly in practice with little tuning. Additionally, for the special case of two objectives, we can leverage the analytical solution to achieve further performance improvements. We re-state Problem $P_A$ for convenience bellow:

$$\min_{\epsilon_1, \dots \epsilon_M} \left\{ \left\| \sum_{m=1}^M \epsilon_m \nabla_\theta \, \mathbb{E}_{d^{\pi_\theta}}[r_m(s,a)] \right\|_2^2 \text{ s.t } \sum_{m=1}^M \epsilon_m = 1, \; \epsilon_m \geq 0, \; \forall m \right\}. \tag{$P_A$}$$

For $M = 2$, the problem reduces to $\min_{\epsilon \in [0,1]} \| \epsilon \nabla_\theta \, \mathbb{E}_{d^{\pi_\theta}}[r_1(s,a)] + (1-\epsilon) \nabla_\theta \, \mathbb{E}_{d^{\pi_\theta}}[r_2(s,a)] \|_2^2$, which is a one-dimensional quadratic optimization problem of a single variable and has the following analytical solution:

$$\epsilon^* = \left[ \frac{(\nabla_\theta \, \mathbb{E}_{d^{\pi_\theta}}[r_2(s,a)] - \nabla_\theta \, \mathbb{E}_{d^{\pi_\theta}}[r_1(s,a)])^T \nabla_\theta \, \mathbb{E}_{d^{\pi_\theta}}[r_2(s,a)]}{\| \nabla_\theta \, \mathbb{E}_{d^{\pi_\theta}}[r_2(s,a)] - \nabla_\theta \, \mathbb{E}_{d^{\pi_\theta}}[r_1(s,a)] \|_2^2} \right]_+, \tag{14}$$

where $[\cdot]_+$ represents the clipping operator, i.e., $[x]_+ = \max(\min(x,1),0)$. Geometrically, Eq. equation 14 implies that minimal-norm point lies either on the boundary on the convex hull ($\epsilon = 0$ or $\epsilon = 1$) and the desired descent direction is parallel to corresponding gradient or is equal to the vector perpendicular to the difference of the gradients. The former one is a corner case and occurs when the ascent direction for one of the objectives is a non-decreasing direction for the other one. While this is only applicable to $M = 2$, it enables us to replace the inner projected gradient descent optimizer in Algorithm 1 with the analytical solution in Eq. equation 14. Additionally, this special case is applicable to several real-world cases since there are several problems that have the *reward-cost* structure.

## B EXPERIMENTAL DETAILS

### B.1 ENVIRONMENTS

Our implementation uses modified versions of 4 OpenAI Gym enviroments (Fig. 5). The first two are synthetic domains and act as a proof-of-concept for our method, while the other two are complex, real domains. The details are listed below:

**Multi-Objective GridWorld (MOGW):** A multi-objective variant of the classic gridworld environment. The agent starts from the top-left corner, moves in a $16 \times 16$-grid and receives a *reward* equal to 1 when it reaches the bottom-right corner. Each action results in a small *time penalty* of $-0.1$. Any policy that results in a trajectory that minimizes the taxicab distance between the agent's starting position and the goal is Pareto optimal. Our implementation uses the `gym-minigrid`[1] enviroment, which we modify to output the aforementioned 2-dimensional reward.

---

[1] https://github.com/maximecb/gym-minigrid

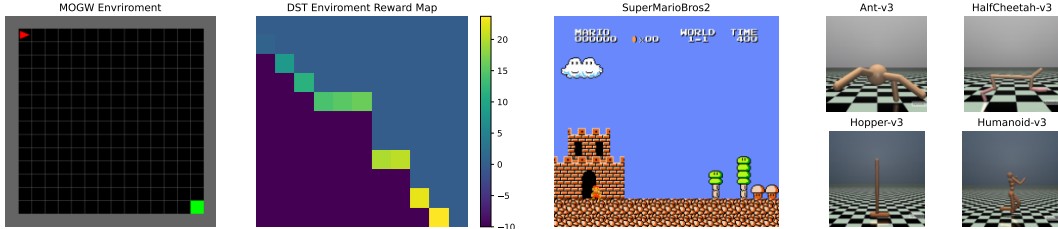

Figure 5: OpenAI Gym enviroments used in our experimental results.

**Deep Sea Treasure (DST)**: A classic multi-objective reinforcement learning environment. The agent is located in a $10 \times 11$-grid and controls a submarine searching for treasures. There are multiple treasures the *value* of which increases as their distance increases. The agent receives a *time penalty* of $-1$ for each action. The Pareto front of non-dominated policies can be easily extracted (51) and is also shown on Fig. 1. As in the MOGW case, we use the `gym-minigrid` package and extend it to the reward map shown in Fig. 5.

**Multi-Objective SuperMario (MOSM)**: We use a modified, multi-objective variant of the popular video game. There are 5 scalar rewards: *x-pos*: the difference in Mario's horizontal position between current and last time point, *time*: a small time penalty of $-0.1$, *deaths*: a large negative penalty of $-100$ each time Mario dies, *coin*: rewards for collecting coins, and *enemy*: rewards for eliminating an enemy. We use the `gym-super-mario-bros`[2] package and the multi-objective extension developed in (26).

**Multi-Objective MuJoCo (MOMU)**: We focus on locomotion tasks and modify the environments provided by the MuJoCo physics simulator to output vector rewards. To do so, we create a custom wrapper for the built-in OpenAI Gym interface. There are 2 scalar rewards: *x-pos*: the difference in the robot's current and last horizontal position, *r-pos*: a positive reward for moving in circular trajectories.

### B.2 IMPLEMENTATION DETAILS

We implement our algorithm in PyTorch building on top of the `torch-ac` package[3]. Our implementation extends the Proximal Policy Optimization (PPO) algorithm to handle our PPA loss function (Eq. equation 6). We use the clipped version of PPO with the clip ratio threshold set to 0.2. We set the GAE parameter to $\lambda = 0.95$ and the discount factor to $\gamma = 0.99$.

**Network Architecture**: We use a consistent network architecture across all our simulation for both the actor and the critic: Two hidden layers of 64 and 64 neurons each and a ReLU non-linearity. In the discreet action case (MOGW, DST, MOSM), we append the actor network with an output layer of dimension equal to the number of actions. In the MuJoCo enviroments, where the actions are continous, we use that output layer to parametrize a multi-dimensional Gaussian distribution, via its mean and variance, assuming uncorrelated actions. Each non-terminal layer has a ReLU non-linearity. Even though the hyperbolic tangent is known to be preferred for reinforcement learning applications, we experimentally observed that it leads to vanishing gradients and slows down learning. This may be due to the fact that the PPA loss function requires second-differentiation. In the SuperMario enviroment, we worked on the pixel domain and two convolutional layer of 32 and 64 layers as feature extractors, which were shared between the actor and the critic.

**Hyperparameters and Training**: To train the neural network, we use the Adam optimizer ($\beta_1 = 0.9, \beta_2 = 0.999$) with a learning rate of 0.0001. In the MOGW, DST and MOMU domains, we optimize the parameters of the actor and critic separately. In the MOSM domain, given that the actor and critic share the convolutional layers, we sum their losses (using weights 1 and 0.5, respectively) and use a common optimizer. We run all of our simulations in the Google Cloud Platorm using 48 vCores and one NVIDIA Tesla T4 GPU. For the MOGW and DST environments, we train the agent for 0.5M frames and for the MOSM and MOMU for 1M frames. For each 512 frames we perform one update of the network parameters iterating over 10 epochs of the collected data and using a mini-batch size of 64.

---

[2] https://github.com/Kautenja/gym-super-mario-bros
[3] https://github.com/lcswillems/torch-ac

