# OpenReview forum: "Pareto Policy Adaptation"
_ICLR.cc/2022/Conference — ICLR 2022 Poster_

### Official Review · Reviewer_hSRE · 2021-10-16

**Correctness:** 3
**Technical Novelty And Significance:** 3
**Empirical Novelty And Significance:** 3
**Recommendation:** 5
**Confidence:** 4

**Main Review:**

The paper is well written and the method is well presented. The introduction covers well prior work. However, I think that the authors do not properly compare their method to the existing ones discussed in the introduction, and I find it hard to distinguish between what is really novel and what has already been done. Furthermore, there are some other issues that should be clarified.

1) The idea of convex coverage set (CCS) is well known in MORL since years. Roijers et al. have published several papers on the subject, e.g., "A Survey of Multi-Objective Sequential Decision-Making" (2013) and "Computing Convex Coverage Sets for Faster Multi-objective Coordination" (2015). Therefore everything up to line 150 is already known (correct me if I am wrong). The novel part should be the gradient descent leveraging on the fist-order condition, resulting in a method that can be applied to a large variety of RL algorithms. I think it would be better to put the CCS part in "Preliminaries", and keep the novel part in Section 3 to highlight it.

2) The first-order condition is a necessary but not sufficient condition for Pareto optimality. You discuss this but only at the very end, as well as the other limitation (PPA can find only convex frontiers). It would be better to write a short paragraph at the end of Section 4 to highlight this. Similarly, you should mention that the Deep Sea Treasure version used for the experiments is not the original version [1]. The original has a convex frontier, which PPA cannot solve.

[1] Vamplew et al., "Empirical evaluation methods for multiobjective reinforcement learning algorithms" (2011)

3) Third, the adaptation part is highly related to Envelope Q-Learning, which you use as baseline in the experiments. You should discuss similarities and differences more in-depth, given that both PPA and EQL retrieve unknown preferences given a solution, and their losses are very similar. For instance, the first term of your Eq. 8 is the same of EQL Eq. 6, while the second term differs from EQL Eq. 7 (I am referring to EQL equations in its paper).

4) It is a bit misleading to claim in the abstract that the final PPA method learns under “unknown preferences” since the objective in (8) does require $\omega$ to evaluate the loss.

5) In Figure 2(b), why PPA is guaranteed to converge to in this example?

6) For the MuJoCo experiments, the paper only considers the more basic baselines (nA-PPA and fixed-preference agents). It would be helpful to compare PPA with other stronger baselines, such as PGMORL (Xu et al., ICML 2020).

7) You should discuss PPA limitation more in details. E.g., PPA directly maximizes the expected return for a given preference or a preference distribution, which means when the environmental preference changes, one has to finetune the model with some extra cost. However, other value-based baselines align the optimal policies with preferences during a single training procedure and can respond with the corresponding optimal policy to any given preference without fine-tuning.

Finally, some questions regarding the experiments.

* Why did you not use the hypervolume as evaluation criterion? It is probably the most common and accurate criterion in MORL literature and could easily replace UT. UT is computed over some samples preferences (how many?), and you could use the same preferences to compute the hypervolume of the corresponding solutions. For MOSM you could use a Monte-Carlo approximation of the hypervolume.
* Why did you not compute PD and UT against baselines for MuJoCo experiments as well?
* Why did you choose to reward the agent for moving in circular paths as second objective for MuJoCo? Usually, the first reward is the distance from a goal, while the second is the energy consumption (see Abdomaleki et al., "A Distributional View on Multi-Objective Policy Optimization" (2020)).
* I am a bit confused by the results. In Table 1, PPA does well under PD and UT criterion. In the plots, results are mixed. PPA does well in MOGW, but is not the best in DST. Then in MuJoCo experiments plots are very noisy, and performance curves sometimes increase and decrease multiple times. For instance, why does w=0.5 increase and then decrease in Hopper? The same curve then performs terribly in Cheetah, why? And why w=0.25 show opposite behavior (decent on Cheetah, terrible in Hopper and Ant)? I am not convinced by your intuition that "given the complex inter-dependence between the two rewards, simply hand-crafting a preference vector is far from optimal". The rewards are the same for all domains, why would Cheetah be different?


**Summary Of The Paper:**

The paper presents a new method for solving MORL problems. The main contributions are: a generic gradient descent MORL algorithm for finding multiple solutions, and another algorithm to retrieves preferences from the given solutions. The authors also present theoretical proofs of their method.

**Summary Of The Review:**

The idea of the paper is interesting, but there are many things that the authors should clarify, both in regards to limitations of the methods and evaluation (metrics, domains, comparison against other baselines).
For these reasons, I vote for its rejection.

---

> ### Author Response · Authors · 2021-11-21
> **Response to reviewer**
>
> > In Figure 2(b), why PPA is guaranteed to converge to in this example?
>
> If we minimize the distance between $\omega^*(\theta)$ and $\omega^0$, then $\omega^0 = \omega^*(\theta)$. This trivial example is meant as an illustrative motivation for introducing the second term in the PPA loss function. The idea is that if we have only one given preference vector $\omega^0$ and we manage to steer (or match) the solution of the PGD solver to that $\omega^0$, then the PPA loss function reduces to the loss function for the scalarized reward (with $\omega^0$ being the scalarization coefficient) and also the ascent steps along its gradient lead to concurrent improvements of all objectives.
>
> > For the MuJoCo experiments, the paper only considers the more basic baselines (nA-PPA and fixed-preference agents). It would be helpful to compare PPA with other stronger baselines, such as PGMORL (Xu et al., ICML 2020).
>
> We added experiments to compare against PGMORL. Please have a look at Table 1.
>
> > You should discuss PPA limitation more in detail. E.g., PPA directly maximizes the expected return for a given preference or a preference distribution, which means when the environmental preference changes, one has to finetune the model with some extra cost. However, other value-based baselines align the optimal policies with preferences during a single training procedure and can respond with the corresponding optimal policy to any given preference without fine-tuning.
>
> We thank you very much for this observation. In the newly added paragraph at the end of Section 4, we discuss this limitation.
>
> > Why did you not use the hypervolume as evaluation criterion? It is probably the most common and accurate criterion in MORL literature and could easily replace UT. UT is computed over some samples preferences (how many?), and you could use the same preferences to compute the hypervolume of the corresponding solutions. For MOSM you could use a Monte-Carlo approximation of the hypervolume.
>
> We thank the reviewer for the suggestion. The hypervolume is indeed the most common criterion in MORL. We calculated it and put the results on Table 1. Additionally, we highlighted the number of preference samples that we used (1000 for the 2-dimensional rewards and 100 for the MultiMario case).
>
> > Why did you not compute PD and UT against baselines for MuJoCo experiments as well?
>
> We computed the metrics against the baselines for the MuJoCo environments and put them on Table 1.
>
> > I am a bit confused by the results. In Table 1, PPA does well under PD and UT criterion. In the plots, results are mixed. PPA does well in MOGW, but is not the best in DST. Then in MuJoCo experiments plots are very noisy, and performance curves sometimes increase and decrease multiple times. For instance, why does $w=0.5$ increase and then decrease in Hopper? The same curve then performs terribly in Cheetah, why? And why $w=0.25$ show opposite behavior (decent on Cheetah, terrible in Hopper and Ant)? I am not convinced by your intuition that "given the complex inter-dependence between the two rewards, simply hand-crafting a preference vector is far from optimal". The rewards are the same for all domains, why would Cheetah be different?
>
> Note that Table 1 compares PPA against other MORL methods using the PD and UT metric, whereas Figure 3 compares it against the fixed-preference variant using the reward curves. While we can see your reasoning (i.e., since it's the best on the former, it should be the best on the later), we can unfortunately not deduce that in practice since we are comparing against different methods and using different metrics.
>
> It is not uncommon for learning curves to increase and decrease multiple times, especially in the MuJoCo environments which are known to be very challenging. We invite you to have a look at https://arxiv.org/pdf/1805.11074.pdf on page 8, where there are two objectives (technically one objective and one constraint): the return and the average torque. You can see that the learning curves are quite noisy and that they increase and decrease multiple times. In addition to that, you will see that different parameter lead to radically different behavior. For instance, for the value $\lambda=100$, the torque in the Humanoid environment does not satisfy the constraint, whereas in the HalfCheetah case it does. Perhaps the intuitive explanation we gave in the paper is not the best one, and it should be replaced or complemented by the fact that the MuJoCo environments are very challenging and the same parameters may lead to radically different results from one environment to another.

---

> > ### Comment · Reviewer_hSRE · 2021-11-29
> > **Most questions have been answered but one remains**
> >
> > Thank you for your reply, and apologies for my late response.
> > You have replied to all my questions but the one about the choice of the rewards in MuJoCo.
> > As already discussed, other SOTA papers use the energy consumption as second reward, while you used 'moving in circular paths'.
> > Why?
> > May it be that this reward is not really conflicting with the first and/or meaningful? This may also explain the weird results in the plot. In your reply about the plots, you also admit that "the intuitive explanation we gave in the paper is not the best one", and it seems that you are also uncertain about the true explanation.
> >
> > Having not fully clear results is a strong limitation, and for this reason --despite I appreciate the other answers and I have increased my score-- I still cannot fully recommend to accept the paper.

---

> ### Author Response · Authors · 2021-11-21
> **Response to reviewer**
>
> We would like to thank you for your time and detailed feedback. Please find our responses to your concerns below. For your convenience, we have highlighted all major changes in the paper.
>
> > The idea of convex coverage set (CCS) is well known in MORL since years. Roijers et al. have published several papers on the subject, e.g., "A Survey of Multi-Objective Sequential Decision-Making" (2013) and "Computing Convex Coverage Sets for Faster Multi-objective Coordination" (2015). Therefore everything up to line 150 is already known (correct me if I am wrong). The novel part should be the gradient descent leveraging on the first-order condition, resulting in a method that can be applied to a large variety of RL algorithms. I think it would be better to put the CCS part in "Preliminaries", and keep the novel part in Section 3 to highlight it.
>
> We thank the reviewer for the suggestion. The majority of information up to line 150 is meant to cover prior work and knowledge. Therefore, we adopted your suggestion and moved that part to the “Preliminaries” section. As a side note, we have not come across the proof of Lemma 1 as given in the paper. However, Lemma 1 is somewhat obvious and generally accepted in the MORL community. Therefore, we moved its proof in the Appendix. Finally, we removed the part that introduced MDPs to avoid having an excessively long “Preliminaries” section.
>
> > The first-order condition is a necessary but not sufficient condition for Pareto optimality. You discuss this but only at the very end, as well as the other limitation (PPA can find only convex frontiers). It would be better to write a short paragraph at the end of Section 4 to highlight this. Similarly, you should mention that the Deep Sea Treasure version used for the experiments is not the original version [1]. The original has a convex frontier, which PPA cannot solve.
>
> Correct. Pareto stationarity is a first-order condition and this is indeed a limitation of our method. As you suggested, we added a paragraph in Section 4 to highlight that limitation. Also, we added a note that we are using a modified version of DST at the beginning of Section 5.
>
> > Third, the adaptation part is highly related to Envelope Q-Learning, which you use as baseline in the experiments. You should discuss similarities and differences more in-depth, given that both PPA and EQL retrieve unknown preferences given a solution, and their losses are very similar. For instance, the first term of your Eq. 8 is the same of EQL Eq. 6, while the second term differs from EQL Eq. 7 (I am referring to EQL equations in its paper).
>
> There are indeed similarities and differences between PPA and EQL. We added a relevant discussion at the end of Section 4. In the same paragraph, we also discuss limitations of our work, especially the fact that Pareto stationarity is a necessary and not sufficient condition, as you suggest below.
>
> > It is a bit misleading to claim in the abstract that the final PPA method learns under “unknown preferences” since the objective in (8) does require $\omega$ to evaluate the loss.
>
> Please note that the PPA loss function in Eq. (6) (we assume that you are referring to Eq. (6) not (8)), requires knowledge of the preference *distribution*, not the preferences. This is what motivates us to talk about *unknown preferences*. Please note that the EQL loss function (which, as you observe, is similar to PPA) takes a similar expectation over the (known) preference distribution, yet the authors mention *unknown preferences* in the paper. In both cases, preferences are treated as a random variable and we assume that they follow a given distribution. Whether or not we can say that a random variable is *unknown* given that we assume knowledge of a distribution, is debatable. We prefer to leave it as is because what we are trying to emphasize is that trade-offs among the objectives are not *precisely* known, we only have knowledge of their distribution, or, in complete lack of any knowledge, we assume uniform distribution. If the trade-offs were known, there would be no need for MORL.

---

### Official Review · Reviewer_1Zn3 · 2021-10-25

**Correctness:** 4
**Technical Novelty And Significance:** 3
**Empirical Novelty And Significance:** 3
**Recommendation:** 8
**Confidence:** 2

**Main Review:**

This work is written well and easy to follow. I didn't find major drawbacks in the paper, and overall the theoretical results seem correct (yet not very novel or fundamental). Overall the work seems solid to me. A few items I would like the authors to address:
1. In the proof of Lemma 1, $d^\pi(s,a)$ can be 0. This would make the lemma incorrect I believe.
2. In Thm 1, $E_{\pi_\theta}$ should be $E_{d^{\pi_\theta}}$
3. In Thm 3, $g_m$ and not defined. I'm assuming these are $g_m(\theta) = \nabla_\theta E_{d^{\pi_\theta}} r_m(s,a)$.
4. The authors wrote: "We start by interpreting the optimal solution of Problem PA as a function of the policy parameters $\omega^* = \omega^*(\theta)$". Can you elaborate on this? Why should the policy and preference parameters have the same parameters?

**Summary Of The Paper:**

This paper introduces a policy gradient approach to multi objective RL. The authors learn show that an optimal solution for a specific reward vector can be found by minimizing a constrained quadratic optimization problem. They propose to solve this problem using projected gradient descent. Then, the authors offer an efficient solution for optimizing the Pareto front through "Pareto Policy Adaptation" loss function, which trades off learning the Pareto front and the policy. The authors conduct experiments and ablations studies on four benchmarks, validating their method with improvement over current sota approaches for MORL.

**Summary Of The Review:**

See review

---

> ### Author Response · Authors · 2021-11-21
> **Response to reviewer**
>
> We would like to thank the reviewer for taking the time to review the paper and for your kind words. Please find our response to your questions below. All major modifications in the revised paper have been highlighted for your convenience.
>
> > In the proof of Lemma 1, $d^{\pi}(s,a)$  can be 0. This would make the lemma incorrect I believe.
>
> Please note that $d^{\pi}(s,a)$ is the discounted state-action occupation measure under policy $\pi$, i.e., it is a probability distribution, therefore it cannot be identically zero. If you are referring to the fact that we multiply and divide with $d^{\pi}(s,a)$, you are correct, there is something missing from the proof. We need to assume that $d^{\pi^\prime} (s,a) = 0$} wherever $d^{\pi}(s,a)=0$. This trick is similar to importance sampling and is commonly used in off-policy reinforcement learning algorithms. Please note that we moved the proof of Lemma 1 to the Appendix.
>
> > The authors wrote: "We start by interpreting the optimal solution of Problem PA as a function of the policy parameters $\omega^*=\omega^*(\theta)$". Can you elaborate on this? Why should the policy and preference parameters have the same parameters?
>
> The objective of the optimization problem P_A depends on the parameters $\theta$ of the policy. So, the optimal solution will also be dependent on the parameters of the policy.
>
> Finally, we fixed the typos that you highlighted.

---

### Official Review · Reviewer_T2hX · 2021-11-01

**Correctness:** 3
**Technical Novelty And Significance:** 2
**Empirical Novelty And Significance:** 2
**Recommendation:** 6
**Confidence:** 3

**Main Review:**

While the idea is interesting, my main concern is about the novelty and the appropriate reference of previous work. As far as I know, the first-order necessary stationary condition was already used together with REINFORCE in previous work (see [31]). You mentioned [31] in the intro but I think a more detailed comparison is due. In [31], the authors already discussed the existence of a common ascent direction and provided the associated quadratic programming approach for the computation. They also presented gradient-based approaches for the computation of first-order stationary policies. I think a deeper discussion of the differences between your and their work is necessary. Given this, I think it is also necessary to add their algorithms to the empirical comparison.

The domains considered in the experiments are standard but I would have appreciated a more extended comparison. For example, why do you focus on domains with very few objectives (3 domains with 2 objectives and one with 5)?

Finally, there are a few typos in the paper that makes the reading complicated, especially in section 3.
- The definition of the value function $v : |S| \to \mathbb{R}^|S|$ is not precise. You should use the notation $v \in \mathbb{R}^{|S|}$ or $v : S \to \mathbb{R}$. Same for the multi-objective case.
- Why are you considering finite state space and continuous actions?
- Eq. 3 is not clear. In particular, the first equality is not correct. There is an expectation w.r.t. the initial distribution missing and probably an additional $w^T$. Please clarify it.
- In the proof of lemma 1, you refer to definition 2. It should be Def. 1
- Based on what you mentioned at the end of section 2.2, there is a constraint missing in definition 2 ($w$ should belong to the simplex).

**Summary Of The Paper:**

The paper presents a multi-objective method for linear preferences. The key concept used for the design of the approach is that there always exists a common ascent direction for all the objectives. The authors leverage this property for introducing a novel loss function that can be integrated into different DeepRL methods. The proposed approach is simple and proved to be effective in the proposed experiments.

**Summary Of The Review:**

My score is justified by the doubts I have about the novelty, comparison with previous work, and clarity of the writing.

---

> ### Author Response · Authors · 2021-11-21
> **Response to reviewer**
>
>
> > The domains considered in the experiments are standard but I would have appreciated a more extended comparison. For example, why do you focus on domains with very few objectives (3 domains with 2 objectives and one with 5)?
>
> We focused on domains that are fairly standardized for evaluating RL agents to facilitate comparison with other methods. The objectives we consider as motivated by practical applications, i.e., (reward, time penalty) or (reward, energy consumption).  Indeed, they are of low dimension. We do not expect the results to change substantially if we increase the dimensionality of the reward. The runtime does not increase either (the PGDSolver in Algorithm 1 can be parallelized). We believe that even low-dimensional rewards suffice for evaluating MORL algorithms (as long as there are at least two objectives).
>
> > The definition of the value function $v : |S| \rightarrow R^{|S|}$ is not precise. You should use the notation  $v \in R^{|S|}$ or $v:S \rightarrow R$. Same for the multi-objective case.
>
> Indeed, the first notation is not precise, but we could not find a part in the paper where we use that notation. Note that we have removed the MDP section from the Preliminaries due to space constraints.
>
> > Why are you considering finite state space and continuous actions?
>
> Yes, correct, it’s not common. We fixed it and assumed that both are discrete. Either way, it does not make much difference, the method is applicable in all cases.
>
> > Eq. 3 is not clear. In particular, the first equality is not correct. There is an expectation w.r.t. the initial distribution missing and probably an additional $\omega^T$. Please clarify it.
>
> Thank you for the suggestion. The $\omega^T$ is indeed missing. We corrected the equation. The expectation with respect to the initial distribution is indeed missing as well. We make the assumption that there exists a given initial state, for the sake of simplicity, to declutter notation. We added that note in the first paragraph of Sec. 2. Please note that we transferred the proof of Lemma 1 to the Appendix, due to space restrictions.
>
> > In the proof of lemma 1, you refer to definition 2. It should be Def. 1
>
> Correct. We fixed it.
>
> > Based on what you mentioned at the end of section 2.2, there is a constraint missing in definition 2 ($\omega^T$ should belong to the simplex).
>
> Correct. We added that constraint.

---

> > ### Comment · Reviewer_T2hX · 2021-11-23
> > **Re: Response to reviewer**
> >
> > Thank you for the detailed answers and for addressing my concerns. Based on this, I've decided to increase my score.

---

> ### Author Response · Authors · 2021-11-21
> **Response to reviewer**
>
> We would like to thank you for taking the time to review our paper. Please find our responses to your questions below. For your convenience, we have highlighted all major changes in our paper.
>
> > While the idea is interesting, my main concern is about the novelty and the appropriate reference of previous work. As far as I know, the first-order necessary stationary condition was already used together with REINFORCE in previous work (see [31]). You mentioned [31] in the intro but I think a more detailed comparison is due. In [31], the authors already discussed the existence of a common ascent direction and provided the associated quadratic programming approach for the computation. They also presented gradient-based approaches for the computation of first-order stationary policies. I think a deeper discussion of the differences between your and their work is necessary. Given this, I think it is also necessary to add their algorithms to the empirical comparison.
>
> We would like to thank you for pointing out that paper. We were aware of it (as well as that line of work on MORL), but admittedly, we had not read ref. [31] thoroughly enough to notice the similarities.  Our method was inspired by the ref. [13] and [35]. The former one is a 2012 paper that introduces the theoretical background (namely, Pareto stationarity and the Multiple-Gradient Descent Algorithm) and the latter one is a 2018 paper that applies the former one on the field of multi-task learning. We did not expect and we are pleasantly surprised to see that such an early MORL paper leverages the idea of common ascent directions to propose a policy gradient algorithm.
>
> We agree with you: The quadratic optimization problems that search for common ascent directions are very similar. The authors in [31] leverage the identified ascent directions to guide gradient-based algorithms towards the Pareto front and correctly state that “each direction in the ascent simplex intrinsically defines a preference over the objectives”. On the other hand, we leverage the common ascent directions to define the PPA loss function, which is a very simple way
> 1. to incorporate the common ascent directions into any learning algorithm and
> 2. to account for a given preference distribution.
>
> Both of the above are enabled due to the use of implicit differentiation, which allows us to backpropagate the gradient of PPA bypassing the operations of the projected gradient descent optimizer that solves the quadratic optimization problem. We believe that the simplicity of PPA is a major advantage of our method: any modern policy gradient algorithm can be tailored to handle multi-objective rewards and preference distributions by a mere modification of the loss function and by implementing a simple projected gradient descent solver. On the practical side, we modified Proximal Policy Optimization to handle our loss function and we showed applications on continuous, high-dimensional control tasks.
>
> Based on the above, we hope that you agree that our method still retains a substantial degree of novelty. We acknowledged the similarities between our method and [31] in a short paragraph at the end of section 1.
>
> Finally, we added the method in [31] to the experimental section. We asked the author of the paper for the code. The code was an old Matlab implementation and, unfortunately, it does not work on the SuperMario and MuJoCo domains due to the lack of environment bindings. We did however include results for the GridWorld and DST environments.

---

### Decision · Program_Chairs · 2022-01-20

**Decision:**

Accept (Poster)

**Comment:**

While this paper has divergent reviews, reviewer hSRE has by far the most detailed review, seems clearly the most informed on the subject, and is the least supportive.  The main issues with the paper seems to be the degree of novelty and reviewer hSRE's feeling that the results on MojuCo are unclear and not explained in the paper.  But this alone does not seem like an adequate reason for rejection and hSRE seems happy with the other aspects of the paper.  Some of hSRE's original complaints do not concern me, such as the fact that first-order stationarity does not imply Pareto-optimality.  I am recommending a poster.